# Application of Network Analysis to Uncover Variables Contributing to Functional Recovery after Stroke

**DOI:** 10.3390/brainsci12081065

**Published:** 2022-08-11

**Authors:** Xiao Xi, Qianfeng Li, Lisa J. Wood, Eliezer Bose, Xi Zeng, Jun Wang, Xun Luo, Qing Mei Wang

**Affiliations:** 1Stroke Biological Recovery Laboratory, Department of Physical Medicine and Rehabilitation, Spaulding Rehabilitation Hospital, The Teaching Affiliate of Harvard Medical School, Charlestown, MA 02129, USA; 2Department of Rehabilitation Medicine, The First Affiliated Hospital of Zhengzhou University, Road No. 169 Nanyang Attachment 10, Zhengzhou 450054, China; 3Department of Rehabilitation Medicine, Nan’ao People’s Hospital, 6th Renmin Street, Dapeng New District, Shenzhen 518121, China; 4William F. Connell School of Nursing, Boston College, Boston, MA 02114, USA; 5School of Nursing, MGH Institute of Health Professions, Charlestown, Boston, MA 02114, USA; 6Department of Anatomy, School of Medicine, Shenzhen University, Shenzhen 518060, China; 7Kerry Rehabilitation Medicine Research Institute, Shenzhen 518048, China

**Keywords:** BDNF (brain-derived neurotrophic factor), LOS (length of stay), network analysis, stroke, recovery

## Abstract

To estimate network structures to discover the interrelationships among variables and distinguish the difference between networks. Three hundred and forty-eight stroke patients were enrolled in this retrospective study. A network analysis was used to investigate the association between those variables. A Network Comparison Test was performed to compare the correlation of variables between networks. Three hundred and twenty-five connections were identified, and 22 of these differed significantly between the high- and low-Functional Independence Measurement (FIM) groups. In the high-FIM network structure, brain-derived neurotrophic factor (BDNF) and length of stay (LOS) had associations with other nodes. However, there was no association with BDNF and LOS in the low-FIM network. In addition, the use of amantadine was associated with shorter LOS and lower FIM motor subscores in the high-FIM network, but there was no such connection in the low-FIM network. Centrality indices revealed that amantadine use had high centrality with others in the high-FIM network but not the low-FIM network. Coronary artery disease (CAD) had high centrality in the low-FIM network structure but not the high-FIM network. Network analysis revealed a new correlation of variables associated with stroke recovery. This approach might be a promising method to facilitate the discovery of novel factors important for stroke recovery.

## 1. Introduction

Stroke is the third-leading cause of disability worldwide, and the number of stroke survivors has significantly increased, leading to a higher burden in the past two decades [1]. Finding variables affecting the functional outcome is essential when designing individualized treatment plans to promote stroke recovery. Many factors, including age, gender, diabetes, coronary artery disease (CAD), stroke volume, and electroencephalography (EEG) have been investigated as predictive variables for stroke outcomes [2,3]. Many of these studies use linear or logistic regression approaches, primary methods for outcome prediction because of the interpretability and ease of utility. However, regression-based methods utilize isolated variables for prediction and analyze the connection between the independent and outcome variables. The performance of the regression approach is affected by the collinearity of the input variables, and therefore is limited by its low power in modeling complex interrelationships between covariates [4].

In order to overcome the above challenges, network analysis has emerged as a promising approach to model complex forms between variables. It has the advantage of machine learning with fewer assumptions about data distribution and can reflect complex interrelationships between predictors and outcomes. This methodology is an aspect of graph theory and has been widely used in various fields, such as lung cancer [5], psychiatry [6,7,8], and climate change [9]. The approach assumes that all variables are constitutive of a disorder so that the connective relationships among these variates initiate and maintain the disorder but avoid the influence of the collinearity of the input variables [10]. The nodes can be any entity or variable, and the edges can be any type of connection [11]. Usually, centrality indices identify important variables of a complicated network. Further, Network Comparison Test (NCT) analysis distinguishes the difference between networks after computing several networks [12]. The strength of this approach gives a unique opportunity to estimate associations between several variables and to recognize how different variables relate to each other. To our knowledge, this analysis approach has not yet been used in patients with stroke.

In this retrospective study, the aim was to estimate two network structures in stroke patients with high- and low-Functional Independence Measurement (FIM) motor function to discover the interrelationships among variables and to investigate the difference between the two networks. The use of amantadine was included to explore the therapeutic benefit of amantadine for stroke patients since amantadine has been proven to be effective for traumatic brain injury (TBI). Brain-derived neurotrophic factor (BDNF) is a recognized marker of neuroplasticity, and as such was included as a potential marker of stroke recovery. We hypothesized that network analysis would allow the identification of factors that are associated with stroke recovery. 

## 2. Materials and Methods

### 2.1. Participants

The study was approved by the Institutional Review Board (IRB). We used our previously published dataset for data analysis [13]. Patients admitted to an acute inpatient rehabilitation hospital from March 2014 to June 2015 were selected using the following inclusion and exclusion criteria. Inclusion criteria consisted of >18 years of age, stroke lesion confirmed by computed tomography (CT) or/and magnetic resonance imaging (MRI), >1 week LOS at the hospital, and peripheral serum samples collected and stored on admission. Exclusion criteria were participants with any missing data. A total of 348 stroke patients met the criteria for inclusion in the study. 

### 2.2. Procedure

Demographic and clinical characteristics were extracted from medical records (Table 1) and included gender, ethnicity, marriage, age, body mass index (BMI), stroke risk factors (stroke history, hypertension, atrial fibrillation, CAD, and diabetes), stroke side, stroke site, type of stroke, aphasia, spasticity, use of amantadine, LOS, discharge destination, FIM motor and cognition subscores on admission and discharge, levels of BUN, creatinine, and hematocrit (HCT). FIM scores were measured after admission and before discharge. Serum BDNF level were measured in duplicate using an enzyme-linked immunosorbent assay (ELISA) (R&D System, Inc., Minneapolis, MN, USA). BDNF measurement was part of this research study and was not a routine clinical test.

### 2.3. Statistical Analysis 

Participants were grouped into a high-FIM motor subscore and low-FIM network structure using the median value, 29, of FIM motor score on admission as a cut-off. Network analysis contains three main steps: (1) estimation of a statistical model; therefore, some parameters from the weighted network can be used to represent a weighted network, (2) analysis of the weighted network based on graph theory, and (3) evaluation of the accuracy of the network (11). Furthermore, we employed NCT to distinguish the differences between estimated networks. R program (3.6.1, A language and environment for statistical computing. R Foundation for Statistical Computing, Vienna, Austria. URL https://www.R-project.org/, accessed on 4 August 2022) and *bootnet* (1.2.3, form Sacha Epskamp), *qgraph* (1.6.3, from Sacha Epskamp)*, ggplot2* (3.2.0, from Hadley Wickham)*, NetworkComparisonTest* (2.2.1, from Claudia van Borkulo) packages were used to conduct this analysis.

#### 2.3.1. Estimating Networks

The Gaussian graphical model was used to estimate the networks. The estimateNetwork function from the *bootnet* package was applied to detect ordinal variables, compute polychoric (or, if needed, polyserial and Pearson) correlations, and estimate network structures automatically. Then, we applied the plot function from *qgraph* to display the network [14]. In the network graph, the width of the edges correlates with the strength of the connections; the blue or red edges indicate positive or negative relationships, respectively.

#### 2.3.2. Computing Centrality Indices

We computed centrality using the centralityPlot function from *qgraph.* For the augments of the plot, *closeness* was the mean length of connected edges, indicating the likelihood that each given node impacts the whole network structure. *Betweenness* was the value of how many times the node was located on the edges between two other nodes, suggesting the contribution of the one node on information flow of the whole network. *Strength* was the sum weight of the connected edges [11,15]. Hence, the large parts of the network may be influenced by altering the nodes with the highest *closeness* and *betweenness*, and many other nodes might be affected by making some adjustments to the node with the highest *strength*. 

#### 2.3.3. Accuracy Test

The stability of networks and centrality indices are quite common after network structures estimated. We used the package *bootnet* to assess the accuracy and used plot function from *qgraph* to display the results. As to the stability plots of the sample size, the lines represented the average correlations between the percent of given samples and the whole sample size, and the shaded area represented the 2.5 and 97.5 percentiles of the estimated samples [16].

#### 2.3.4. Network Comparison

To estimate the structural level differences between networks, we used a newly developed R package *NetworkComparisonTest* from Claudia van Borkulo, et al. [17]. This analysis gave us the Bonferroni corrected *p*-value of edges. A *p* < 0.05 was considered as significantly different. From the results, we concluded which node played a different role between networks. In addition, we used a paired t-test to compute the global strength difference or density difference of connection.

## 3. Results 

In order to estimate the interrelationship of variables, the Gaussian graphical model was applied to compute the high- and low-FIM networks. We used the same data (as shown in Table 1) from our previous publication that included 26 variables of demographic and clinical characteristics, and serum biomarkers [13]. There were 174 participants in each group, and no significant difference was found between the two groups regarding age, gender, BMI, ethnicity, stroke type, stroke side, HCT, BUN, creatinine level, the occurrence of CAD, diabetes, hypertension, atrial fibrillation (AF), and prior stroke. Moreover, significant differences were found in LOS, discharge destination, serum BDNF level, FIM motor subscores on admission (FMA) and discharge (FMD). 

### 3.1. High-FIM Network 

The high-FIM network structure (Figure 1A) showed strong positive connections between BUN and creatinine, and strong negative connections between spasticity and discharge destination (with spasticity more likely on discharge home). Centrality indices results revealed that amantadine use had the highest strength, closeness, and betweenness (Figure 2, Table 2) among all variables analyzed, suggesting the administration of amantadine had the most interactions with other covariants. As shown in Figure 3A, stability was reduced as the sample size decreased. Corstability (CS-coefficient) of strength was 0.05 (cor = 0.7) and was under the cut-off of 0.5, which is considered a required metric stable. 

### 3.2. Low-FIM Network 

As shown in Figure 1B in the low-FIM network, stroke side had a strong negative connection with spasticity and a strong positive connection with stroke site. In addition, strong positive correlations were found in the high-FIM network between FCD and FCA, and BUN and creatinine. The centrality indices plot (Figure 2, blue) and centrality scores (Table 2) revealed that CAD was the most central variable in the low-FIM network. In addition, when the sample size decreased, the strength was unstable (Figure 3B), since the CS-coefficient of strength was (CS (cor = 0.7) = 0.05).

### 3.3. Network Comparison

NCT was performed to quantify the differences in the connection weights to further investigate the overall differences between these two network structures. Twenty-two of 325 connections differed significantly between networks (Table 3). In addition, LOS was linked to five of the 22 connections, implying it was significantly different in the two networks and formed more complicated connections in the high-FIM network. Interestingly, factors that affect LOS were different between the high-FIM and low-FIM networks. In the high-FIM network, high FMA and FMD, as well as being married and female were more likely to show a decreased LOS; however, in the low-FIM network, these factors were not correlated with LOS. The paired t-test revealed that the global strength was significantly different between the two networks (*p* < 0.001), and the high-FIM network had more dense connections. 

## 4. Discussion

In this study, we found that network analysis may be a promising tool for discovering complex interrelationships of variables contributing to stroke recovery. Comparing high-FIM and low-FIM networks revealed that twenty-two out of 325 connections differed significantly between networks. The high-FIM network had more dense connections, with several correlations with potential clinical implications. 

First, node amantadine strongly interacted with other variables in the high-FIM network. The use of amantadine tended to decrease LOS in the high-FIM network structure but not in the low-FIM network structure, suggesting that different motor functions may influence the relationship between amantadine use and LOS. Although much evidence supports amantadine’s use in improving consciousness, cognition, and disability level for patients with traumatic brain injury (TBI) [18,19,20], the effect of amantadine on stroke recovery is less clear [21,22]. As an NMDA receptor (NMDA-R) antagonist, amantadine may also be anti-inflammatory. Indeed, it was reported that amantadine has an inhibitory effect on microglial activation and the signaling pathway [23], implying amantadine use for stroke patients where neuroinflammation is common. 

Second, the BDNF node was positively connected with diabetes in the high-FIM network but not in the low-FIM network. Our findings are consistent with prior studies that did not show a significant association between serum BDNF and diabetes [24,25]. However, other studies have shown such an association [26,27]. This incongruence may be attributed to variations in BMI [26], ethnicity [28]. Many factors can impact circulating BDNF levels, including vigorous aerobic exercise [29], calorie restriction [30,31], blood glucose levels [32], platelet activity, gender, and cognition [33]. Moreover, BDNF has two isoforms (pro-BDNF and mature BDNF) and has two transmembrane signaling pathways through receptors tropomyosin-related kinase B (TrKB) and p75 neurotrophic receptor (p75NTR) [34]. Mature BDNF specifically binds to TrKB promoting cell survival, whereas pro-BDNF preferentially binds to p75NTR resulting in apoptosis. Loss of TrKB signaling is reported in aging and different neurogenerative disorders. Therefore, the functional outcome of BDNF depends on which isoforms of BDNF and the receptors, not just the level of circulating BDNF. Consideration of these other factors will be important in future studies. 

Third, the LOS node was connected to several nodes in the high-FIM network, including FMD (negative), FMA (negative), gender (negative), amantadine (negative), and marriage (positive). However, the LOS node had no connections in the low-FIM network. Our findings are consistent with other studies showing LOS correlated with functional outcomes on admission, gender [35], and marriage status [36]. The lack of an association in the low-FIM, suggests that variables affecting LOS are different in the high-FIM and low-FIM groups. Further study is needed to dissect the contributing factors to LOS that would provide insight into treatment and discharge plans. 

In addition, a strong negative connection was found in the high-FIM network between discharge destination and spasticity, suggesting that patients with spasticity are more likely to be discharged home. This is consistent with a study reporting that stroke patients with spasticity have better outcomes than non-spastic patients [37]. However, other studies report the opposite correlation in chronic stroke [38,39]. This incongruence may be related to different stages of recovery [40]. 

The node CAD has high centrality in the low-FIM network but not in the high-FIM network. CAD may have more apparent interactions with other clinical factors in the low-function group compared to the high-function group. This finding is inconsistent with the finding that CAD is a major detrimental factor for patients with low mobility [41]. Our data suggest that in patients with high mobility, CAD appears to be a less limiting factor. 

The primary strength of the network analysis approach is interrelationship discovery, which may reveal unknown relationships between variables. In this study, some connections, including the correlation between serum BDNF levels with diabetes–stroke patients and amantadine, were not confirmed. 

### Limitations

The cohort consisted of stroke patients from one urban rehabilitation hospital; therefore, conclusions cannot be generalized to the general population. Causal associations cannot be defined in this study [42], and experimental and prospective studies must confirm a causal relationship [43]. Another limitation is the small sample size. Future studies with larger sample sizes are necessary [44]. The infarct volume is an important variable for stroke patients, but it was not included in this study since not all patients had imaging available for analysis. Lacking in our dataset was the length of time between stroke onset and initiation of rehabilitation, which is variable, and likely impacts functional outcomes. 

## 5. Conclusions

The described network analysis approach offers a unique opportunity to learn how demographic and clinical characteristics and serum biomarkers may contribute to stroke recovery. Dissimilar performance of variables has been displayed between high- and low-FIM networks, including LOS, serum BDNF level, and amantadine use. 

## Figures and Tables

**Figure 1 brainsci-12-01065-f001:**
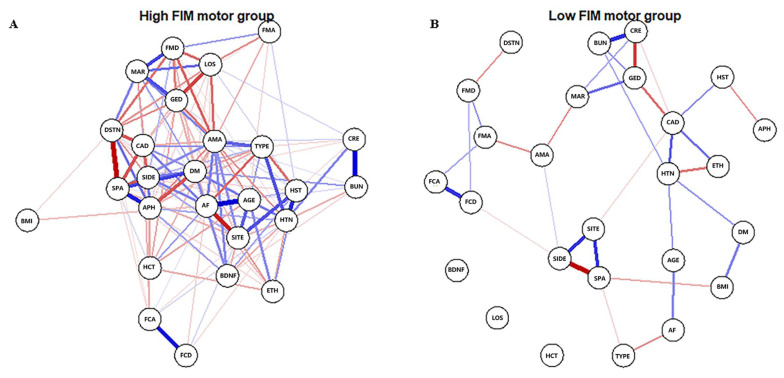
Estimated network structures of high (**A**) and low (**B**) motor function. The weight edges represent the strength of connections, while the blue or red edges stand for positive or negative relationships, respectively. The strongest negative connections were between spasticity and discharge destination in the high-FIM network and between spasticity and stroke side in the low-FIM network.

**Figure 2 brainsci-12-01065-f002:**
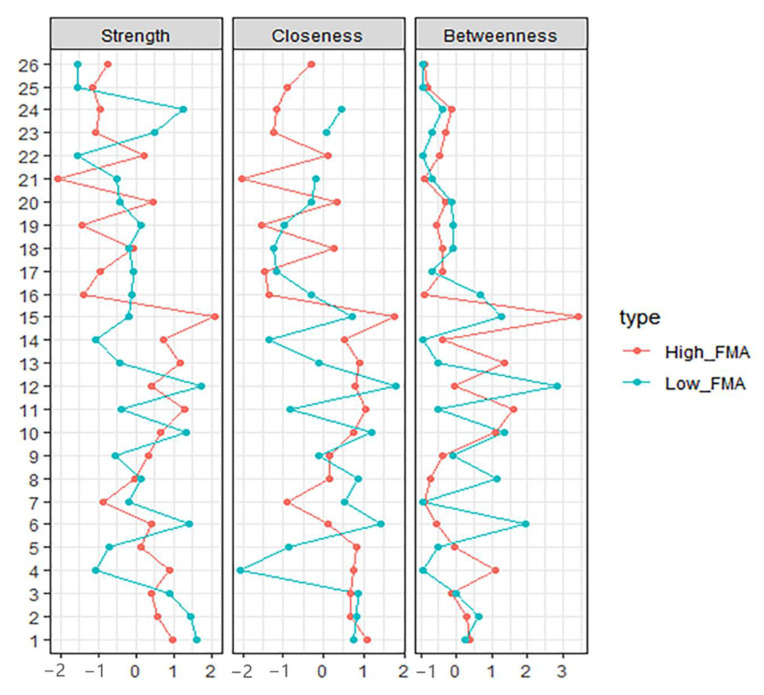
Centrality Indices results of high-FIM network (red) and low-FIM network (blue). Strength represents the quantity of connections, while closeness and betweenness represent the interaction with other nodes.

**Figure 3 brainsci-12-01065-f003:**
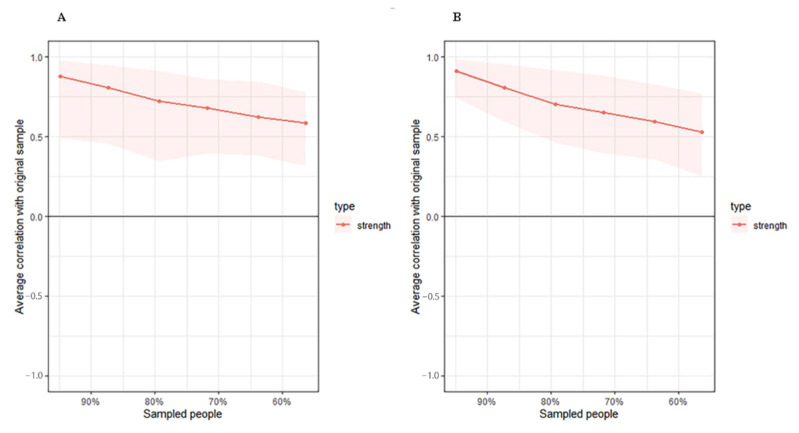
Estimated sample stability of strength in high (**A**) and low (**B**) motor function. Lines represent the average correlation, and areas represent the change interval.

**Table 1 brainsci-12-01065-t001:** List of demographic and clinical characteristics, serum biomarkers, and corresponding node IDs.

ID	Variable	Abbreviation
1	Spasticity (non-spastic vs spastic)	SPA
2	Stroke side (right side vs left side vs both sides)	SIDE
3	Stroke site (supratentorial vs infratentorial vs both sites)	SITE
4	Discharge destination (home vs skilled nursing facility vs acute hospital)	DSTN
5	Stroke type (ischemic stroke vs hemorrhage stroke)	TYPE
6	Gender	GED
7	Ethnicity (Hispanic vs non-Hispanic)	ETH
8	Marriage (married vs unmarried)	MAR
9	Prior stroke history	HST
10	Hypertension	HTN
11	Atrial fibrillation (AF)	AF
12	Coronary artery disease (CAD)	CAD
13	Diabetes mellitus (DM)	DM
14	Aphasia	APH
15	Amantadine	AMA
16	FIM motor subscores on admission (FMA)	FMA
17	FIM cognitive subscores on admission (FCA)	FCA
18	FIM motor subscores at discharge (FMD)	FMD
19	FIM cognitive subscores at discharge (FCD)	FCD
20	Age	AGE
21	Body mass index (BMI)	BMI
22	Length of stay (LOS)	LOS
23	Blood urea nitrogen (BUN)	BUN
24	Creatinine	CRE
25	Hematocrit (HCT)	HCT
26	brain-derived neurotrophic factor (BDNF)	BDNF

**Table 2 brainsci-12-01065-t002:** Differences in Two Groups of Centrality Indices.

ID	Variables	High Motor Function			Low Motor Function		
		Betweenness	Closeness	Strength	Betweenness	Closeness	Strength
1	Spasticity	0.36	1.07	0.98	0.26	0.75	1.60
2	Stroke side	0.28	0.65	−0.58	0.62	0.83	1.44
3	Stroke site	−0.14	0.68	0.40	−0.02	0.87	0.89
4	Destination	1.11	0.75	0.88	−0.93	−2.05	−1.06
5	Stroke type	−0.06	0.84	0.15	−0.54	−0.87	−0.69
6	Gender	−0.56	0.10	0.41	1.93	1.43	1.40
7	Ethnicity	−0.98	−0.91	−0.86	−0.93	0.51	−0.18
8	Marriage	−0.72	0.17	−0.03	1.14	0.86	0.13
9	Stroke history	−0.39	0.13	0.31	−0.10	−0.10	−0.54
10	Hypertension	1.11	0.76	0.66	1.33	1.20	1.35
11	AF	1.61	1.05	1.29	−0.50	−0.81	−0.38
12	CAD	−0.06	0.78	0.42	2.85 §	1.81 §	1.74 §
13	Diabetes	1.36	0.89	1.18	−0.50	−0.12	−0.42
14	Aphasia	−0.39	0.54	0.71	−0.93	−1.35	−1.05
15	Amantadine	3.44 §	1.75 §	2.07 §	1.25	0.70	−0.18
16	FMA	−0.98	−1.36	−1.39	0.66	−0.31	−0.10
17	FCA	−0.39	−1.46	−0.96	−0.69	−1.18	−0.07
18	FMD	−0.39	0.28	−0.05	−0.10	−1.23	−0.20
19	FCD	−0.56	−1.54	−1.42	−0.10	−0.97	0.13
20	Age	−0.31	0.34	0.44	−0.14	−0.31	−0.42
21	BMI	−0.89	−2.01	−2.06	−0.69	−0.18	−0.51
22	LOS	−0.47	0.10	0.22	−0.93	NA †	−1.53
23	BUN	−0.31	−1.25	−1.08	−0.69	0.08	0.48
24	Creatinine	−0.14	−1.16	−0.95	−0.38	0.44	1.25
25	HCT	−0.81	−0.89	−1.13	−0.93	NA †	−1.53
26	BDNF	−0.89	−0.29	−0.76	−0.93	NA †	−1.53

† Three closeness scores were missing due to weak connection with the largest component of the network structure. § Higher scores represent higher centrality. AF: Atrial fibrillation, CAD: Coronary artery disease, FMA: FIM motor subscores on admission, FCA: FIM cognitive subscores on admission, FMD: FIM motor subscores at discharge, FCD: FIM cognitive subscores at discharge, BMI: Body mass index, LOS: Length of stay, BUN: Blood urea nitrogen, HCT: Hematocrit, BDNF: brain-derived neurotrophic factor.

**Table 3 brainsci-12-01065-t003:** Variable correlations that differ significantly between the High-FIM and Low-FIM networks.

Correlation Represented by the Edge of Two Variables	*p* Value
FMA-Marriage	<0.001
FMD-LOS	<0.001
FMD-Gender	<0.001
Marriage-LOS	0.001
Gender-LOS	0.003
Amantadine-LOS	0.011
Age-HCT	0.015
Destination-AF	0.023
Spasticity-Aphasia	0.027
Stroke Site-Age	0.027
Spasticity-Destination	0.028
Amantadine-FMD	0.031
Stroke Site-Stroke History	0.033
Stroke History-Hypertension	0.033
Spasticity-Stroke Site	0.034
Destination-Aphasia	0.035
AF-Age	0.04
Destination-LOS	0.04
AF-HCT	0.042
Marriage-Amantadine	0.044
DM-Aphasia	0.045
Stroke Type-Hypertension	0.046

*p* < 0.05 were considered as significantly different.

## Data Availability

Not applicable.

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
