# Peer review of "Application of Network Analysis to Uncover Variables Contributing to Functional Recovery after Stroke"

_brainsci, 2022, doi:10.3390/brainsci12081065_

Round 1

Reviewer 1 Report

The current manuscript is well written. The aim was to assess network analysis to better assess interrelationship amongst various stroke-associated variables. Several nodes were highlighted from the analysis and I only have a couple of minor commons on a couple of these. 

1) Amantadine, whilst showing some effect on recovery following TBI, this effect is only observed in some studies, as others studies looking at chronic TBI patients have shown Amantadine not to work. A more balanced discussion around this should be included. Authors should also mention receptor signalling systems involved following amantadine treatment, and look towards preclinical studies where more specific targeting to some of these receptor signalling systems have already been addressed. It is also possible that Amantadine may only work if the stroke or TBI is located in specific cortical and subcortical structures.

2) BDNF plays a major role is plasticity and axonal sprouting.  In addition, BDNF decreases with age and this correlates with cognitive decline. It should be mentioned that BDNF can signal through two different receptors the TrKB receptor, which is the predominant receptor, but also the p75NTR, which when activated can stimulate apoptosis and impaired axonal outgrowth. As the TrKB receptor is often down related with age as well as with various pathological condition, it’s worth noting whether it known if TrKB is altered and if the changes positive correlation with BDNF is the pro or mature from of BDNF as this can also impact on which receptor BDNF is acting through.  The take home here is that just because there is an change in BDNF and a positive correlation with diabetes, doesn’t means it’s a good change, as it all depends on which isoform of BDNF and the receptor that its signalling through.

Reviewer 2 Report

Xi et al are presenting the study “application of network analysis to uncover variables contributing to functional recovery after stroke”. It is an original approach in the field, with potential interest. I do have some relevant points to be clarified :

1.The main criticism about this methodological approach is that the study of associations between biological, socio-economic or process of care factors with any health outcome should be supported by a prior logical analysis. Otherwise, the results will be non-informative. For instance, the authors should explain the inclusion of amantadine or brain-derived neurotrophic factor (BDNF) as variables. Are we analyzing the indication for amantadine use?

2. Why was brain-derived neurotrophic  factor (BDNF) measured ? is it a routine test ?

3.“he strength of this approach is in giving a unique opportunity to estimate associations of serious variables and recognize how different variables relate to each  other. To our knowledge” – define “serious variables”

4.“stroke focus confirmed by computed tomography (CT) or/and 82 magnetic resonance imaging (MRI)” – stroke focus or stroke lesion ?

5.Detail how the sampling or inclusion process was made

6.. Include the median /mean time from stroke onset to rehabilitation in the analysis  

Reviewer 3 Report

Review of a manuscript -Manuscript ID: brainsci-1815770

The first thing that caught my attention was that the authors probably did not prepare the manuscript in accordance with the guidelines for the authors. Please read and follow the Brain Sciences Writer's Guidelines.

There should be no headings in the abstract.

The first section is the introduction, not the background. Please correct.

The second section is Materials and Methods, not Methods.

Besides, I see incorrectly presented references in the text. They should be presented in brackets [  ].

There is also a lack of final information, such as Author Contributions.

Please review and follow the authors' guide.

Apart from that,

The paper presents interesting studies important in the treatment of patients after stroke.

The title encourages to read the content of the article.

The research results confirm the assumptions of the work.

This study is important for the prevention and treatment of stroke patients.

Round 2

Reviewer 2 Report

The auhtors responded to all my comments. I do not have any further comments.